# Deep genome sequencing reveals extensive genetic heterogeneity in early human placentas

Ieva Miceikaite [1,2] ✉, Christina Fagerberg [1,2,3], Charlotte Brasch-Andersen [1,2], Pernille M. Torring [2,4], Britta Schlott Kristiansen [2], Qin Hao [2], Lene Sperling[4,5] & Martin J. Larsen [1,2,4] ✉

Placental biopsy in early pregnancy is widely used in prenatal genetic diagnostics as a surrogate for fetal tissue. Confined placental chromosomal mosaicism is a well-documented phenomenon causing genetic discrepancies between the fetus and placenta. Although comprehensive sequencing methods are becoming popular for prenatal screening of monogenic disorders, knowledge of concordance between the fetus and early placenta at the sequence level remains limited. By deep genome sequencing, we have mapped the mutational landscape across multiple sites and stages of placental development. We have revealed wide-spread mutations, with distinct clusters of postzygotic non-fetal small sequence variants, indicating extensive clonal evolution in all early placental biopsies, including first-trimester chorionic villus samples. Our study illuminates spatial and temporal genetic heterogeneity of the developing placenta. While most clonal sequence variants in placental biopsies exhibit low variant allele frequency, their presence underscores the need for caution when using placental tissue as a fetal proxy for diagnostics. These findings highlight the importance of confirmatory testing using AF in cases where placental mosaicism is suspected to avoid misinterpretation and unnecessary interventions.

Comprehensive sequencing is becoming an important diagnostic tool for pregnancies with abnormal ultrasound scan findings[1]. As a supplement to chromosomal analysis, whole exome sequencing (WES) or whole genome sequencing (WGS) of early-stage, uncultured placental biopsies (chorionic villus sample, CVS) is being implemented to identify rare monogenic congenital disorders. The understanding of chromosomal heterogeneity in the placenta and fetal tissues has been well established by extensive research, which has consistently shown that discrepancies between the prenatally investigated placental biopsy and the fetus can lead to misdiagnosis and adverse pregnancy outcomes[2]. For instance, a chromosomal aberration confined to the placenta, i.e., confined placental mosaicism (CPM), may be misinterpreted as a fetal abnormality. Without awareness of this phenomenon, such findings may lead to unnecessary medical interventions, emotional distress, and ill-informed decisions regarding the management of pregnancy[3,4]. CPM for chromosomal aberrations has been reportedly detected in up to 2% of all pregnancies[5], necessitating confirmatory genetic testing on amniotic fluid (AF) when CPM is

[1]Human Genetics Unit, Department of Clinical Research, Faculty of Health Sciences, University of Southern Denmark, Odense, Denmark. [2]Department of Clinical Genetics, Odense University Hospital, Odense, Denmark. [3]Department of Clinical Genetics, Lillebaelt Hospital, Vejle, Denmark. [4]Center for Fetal Genetics, Odense University Hospital, Odense, Denmark. [5]Fetal Medicine Unit, Department of Obstetrics and Gynecology, Odense University Hospital, Odense, Denmark. ✉e-mail: imiceikaite@mgh.harvard.edu; martin.larsen@rsyd.dk

suspected. While previous studies have focused solely on large chromosomal and copy number changes confined to placental tissue[6], recent research employing advanced genomic sequencing technologies, facilitating more comprehensive investigations of placental heterogeneity, has revealed that CPM at the level of small-sequence variants (SSVs) is a much more common characteristic of term placentas than large chromosomal aberrations[7]. However, studying placental mosaicism across gestation is challenging due to ethical and logistical constraints, particularly in healthy ongoing pregnancies, limiting access to sequential placental samples. Therefore, despite its potential clinical significance, fundamental aspects of the presence and extent of postzygotic sequence variants (PZVs) in early placental development remain largely unexplored.

Here, we investigate spatial and temporal placental heterogeneity by analyzing postzygotic changes in the early and late placental genome sequence, focusing on single nucleotide variants (SNVs) or small insertions–deletions (indels)—collectively defined as SSVs throughout this paper. We compare the outcomes of deep genome sequencing of four postgestational biopsies from different placental quadrants against corresponding prenatal modalities (CVS, AF, and/or cell-free DNA (cfDNA) from maternal plasma) and fetal reference samples obtained following delivery or termination. We perform heterogeneity analysis by comparing postzygotic SSVs (PZVs) and chromosomal changes in placental biopsies sampled from different quadrants and at different gestational stages. Additionally, we investigate the prevalence of fetal de novo germline SSVs in the placental samples. Our results reveal widespread placental mosaicism and distinct clonal expansions that grow in size and complexity with advancing gestation, highlighting the need for confirmatory AF testing when sequence-level placental mosaicism is suspected. This detailed investigation of early placental genetic heterogeneity fills a critical gap in our understanding of placental biology at the sequence level, which is essential for developing diagnostic strategies for prenatal detection of monogenic disorders.

## Results

### Deep genome sequencing reveals spatial genetic heterogeneity with multiple unique clones in both early and late stage placentas

We included six pregnancies at different stages of placental development: early second trimester, i.e., GA 14–16 (cases 1–2), mid-late second trimester, i.e., GA 18–22 (cases 3–5), and term placenta, i.e., GA 37 (case 6). For each placenta, we collected post-gestational placental biopsies from four different sites (quadrants) alongside fetal tissue, prenatal CVS or AF samples, and parental blood samples. Detailed characteristics of all the included cases are provided in Table 1. Deep genome sequencing was performed on placental and fetal samples with a mean coverage of 146× (range: 99–213×), allowing the genome-wide PZV analysis (see "Online Methods"). Parental samples were included for the identification of de novo germline SSVs (Supplementary Table 1).

We identified PZVs and chromosomal changes detected solely in placental biopsies and not in the fetuses. The key characteristic observed was that all placental biopsies (24 of 24), irrespective of gestational age (GA), had private clonal expansions, with each sample containing unique clusters of PZVs. These individual clones composed 14–62% of the total cell populations in a given biopsy (Fig. 1). Each clone was defined by a distinct cluster of PZVs identified solely in placental biopsies. Clonal expansion refers to the fraction of the total number of sampled cells in a biopsy specimen carrying a specific clone. All PZVs that defined a unique cluster had similar variant allele frequencies (VAF), showing a close relationship, thus defining a separate private clone for that sample, which was not shared across other placental samples (Supplementary Fig. 1). Figure 1 visually summarizes these finding by displaying the proportion of cells in each biopsy

**Table 1 | Characteristics of cases and samples included in the study**

| Case no. | CVS GA | AF GA | Placenta GA | Fetal tissue GA | cfDNA GA | Indication for genetic testing | Clinical findings | Pregnancy outcome |
|---|---|---|---|---|---|---|---|---|
| 1 | 12+5 | N/A | 14+1 | 14+1 | 13+1 | Hydrops fetalis | Monosomy X | Missed abortion at GA 14+1 (detected GA 16+) |
| 2 | 13+0 | N/A | 15+6 | 15+6 | 14+5 | NT 8.9 mm, suspected heart malformation, diaphragmatic hernia | NR2F2(NM_021005.4): c.1044_1048delCGTTA (de novo) | Terminated at GA 15+6 |
| 3 | 14+0 | N/A | 19+0 | 19+0 | 14+0 | Large abdominal bubble | none | Terminated at GA 19+0 |
| 4 | N/A | 18+3 | 20+2 | 20+2 | 18+3 | Cyst fossa posterior, unilateral cleft lip-gum, suspected heart malformation | 22 Mb deletion on chromosome 2 | Terminated at GA 20+2 |
| 5 | N/A | 20+0 | 21+5 | 21+5 | 21+1 | Hydrocephalus | none | Terminated at GA 21+5 |
| 6 | 12+1 | 16+1 | 37+0 | 37+0[a] | N/A | Increased risk for Trisomy 21 (>1:20) | Trisomy 21, CPM | Live birth (GA 37+0) |

GA gestational age, cfDNA cell-free DNA, NT nuchal translucency, N/A not available/not applicable, CPM confirmed confined placental mosaicism.
[a]Umbilical cord blood sample (in other cases, fetal Achilles tendon).

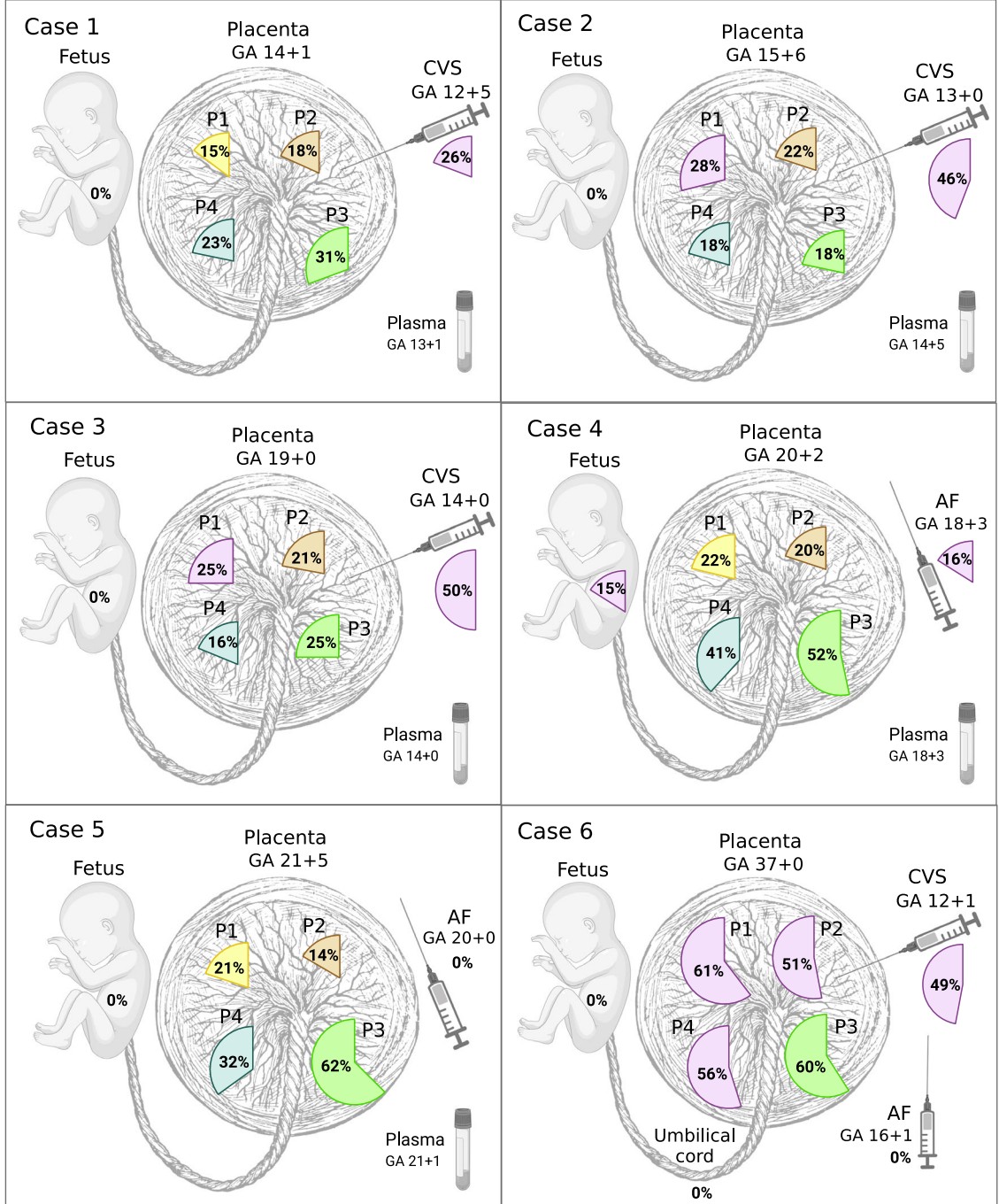

**Fig. 1 | Dominant clonal expansions of postzygotic small sequence variants (PZVs) were identified in various samples for each case.** The percentages indicate the fraction of cells within the largest private clone in each sample. AF amniotic fluid, CVS chorionic villus sample, GA gestational age (in weeks), P1–P4 placental biopsies from different quadrants, UC umbilical cord blood. Created in BioRender. Larsen, M. (2025) https://BioRender.com/d8vsj1x.

represented by the largest clonal expansion, emphasizing the extent of spatial genetic heterogeneity across different placentas. Supplementary Fig. 1 further supports this by presenting VAF jitter plots, illustrating the distinct clusters of PZVs within each sample. This visualization seeks to aid in understanding the heterogeneity and nature of private clonal expansions, which likely arise due to postzygotic events early in placental development. In this study, we defined clones by the clusters of PZVs, which included at least 10 PZVs. Each placental biopsy carried at least one exclusive clone, which was not detected in any other biopsy from the same case (Fig. 2B). For instance, in case 1, which was from the early second trimester, a total of six

unique clones were identified in the placenta biopsies, each consisting of 12–125 PZVs (Fig. 2A). In each of the four placental biopsies, a unique clone was identified, which was found exclusively in that corresponding biopsy. The specific cell clones accounted for 13–31% of the total cell population. Similar patterns were consistently observed in the other cases included in the study, both in second and third-trimester placentas. Unique clones comprising >10% of the total number of cells were detected in all biopsies, without exception. From a clinical perspective, it is important to note that the vast majority of these PZVs are in non-coding regions of the genome. While each case contained a total of 254–998 PZVs (Fig. 2B), only 4–11 of these were in

**A**    Postzygotic small sequence variant clones across different samples for Case 1

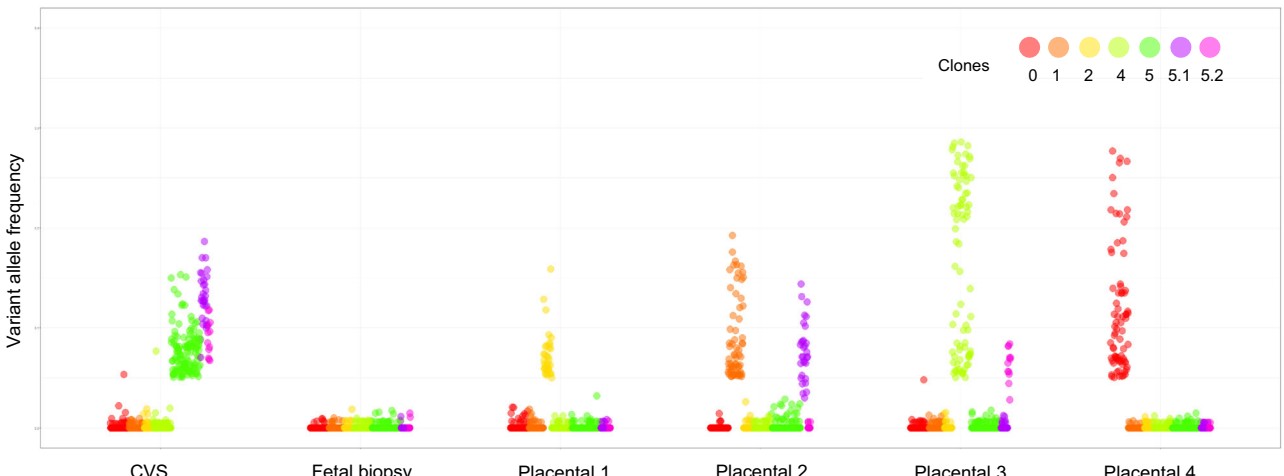

**B**    Clonal expansions and mutational burden identified per sample across case

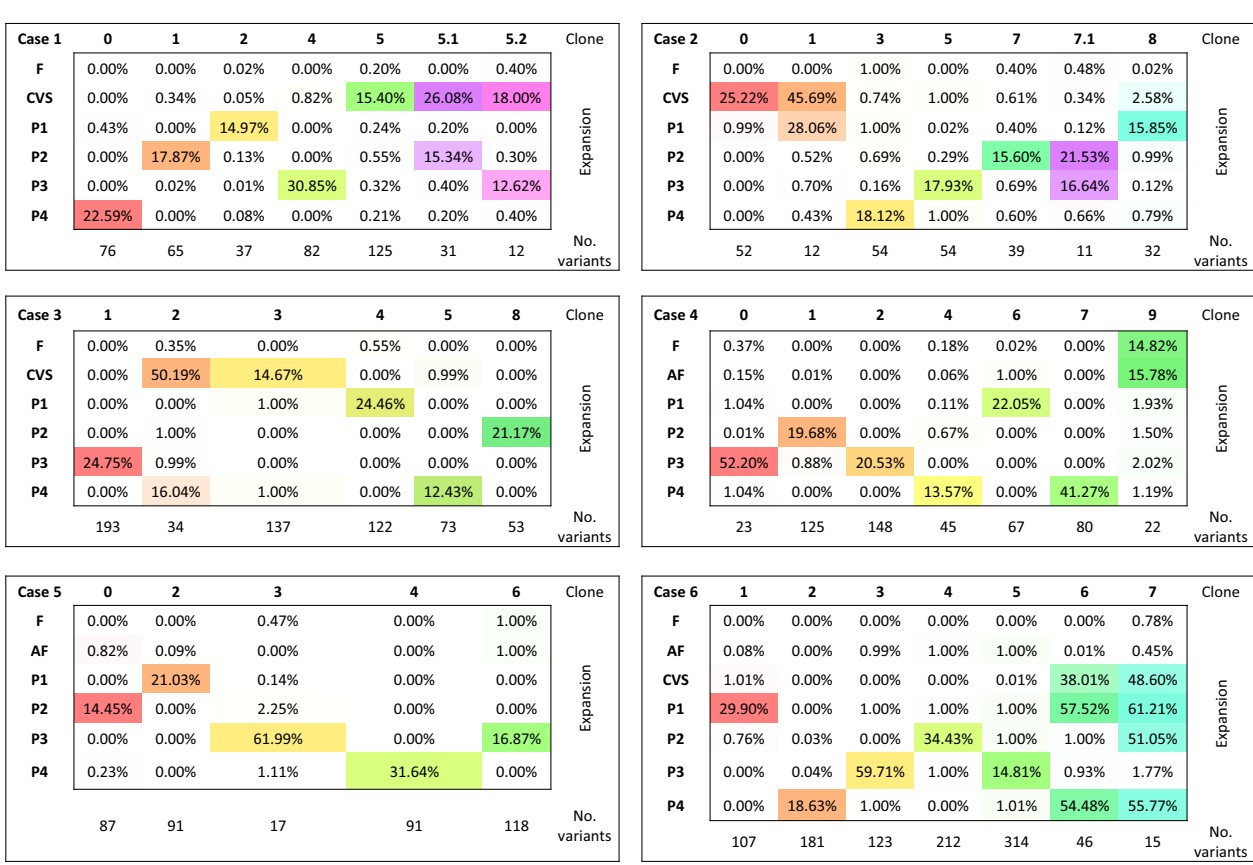

**Fig. 2 | Spatial distribution and clonal structure of postzygotic variants across fetal and placental tissues. A** Variant Allele Frequency (VAF) jitter plot for case 1, depicting postzygotic small sequence variant (PZV) clones across different samples. All cases are shown in Supplementary Fig. 1. **B** Clonal expansions across samples are defined by clusters of PZVs. The table shows the percentage of cells containing each clone, and the lower rows indicate the number of PZVs defining each clone. In case 1, each placental sample harbors a unique private clone. In cases 2 and 3, some placental samples and CVS share overlapping clones, while case 4 reveals potential confined fetal PZVs, detected at low fractions in fetal tissue and amniotic fluid samples. Source data are provided as a Source data file. *AF* amniotic fluid, *CVS* chorionic villus sampling, *F* fetal tissue, *P*1–*P*4 placental biopsies.

the coding regions (Supplementary Table 2). To independently confirm the presence of postzygotic variants identified in the genome analysis, we performed deep targeted sequencing on the P3 biopsy from Case 2, verifying the presence of private clones in placental tissue

by identification of identically somatic variant clusters (Supplementary Fig. 5). In addition, CVS samples from Cases 1–3 were also sequenced independently in a clinical diagnostic setting. We consistently identified the same somatic variant clusters in both the research and clinical

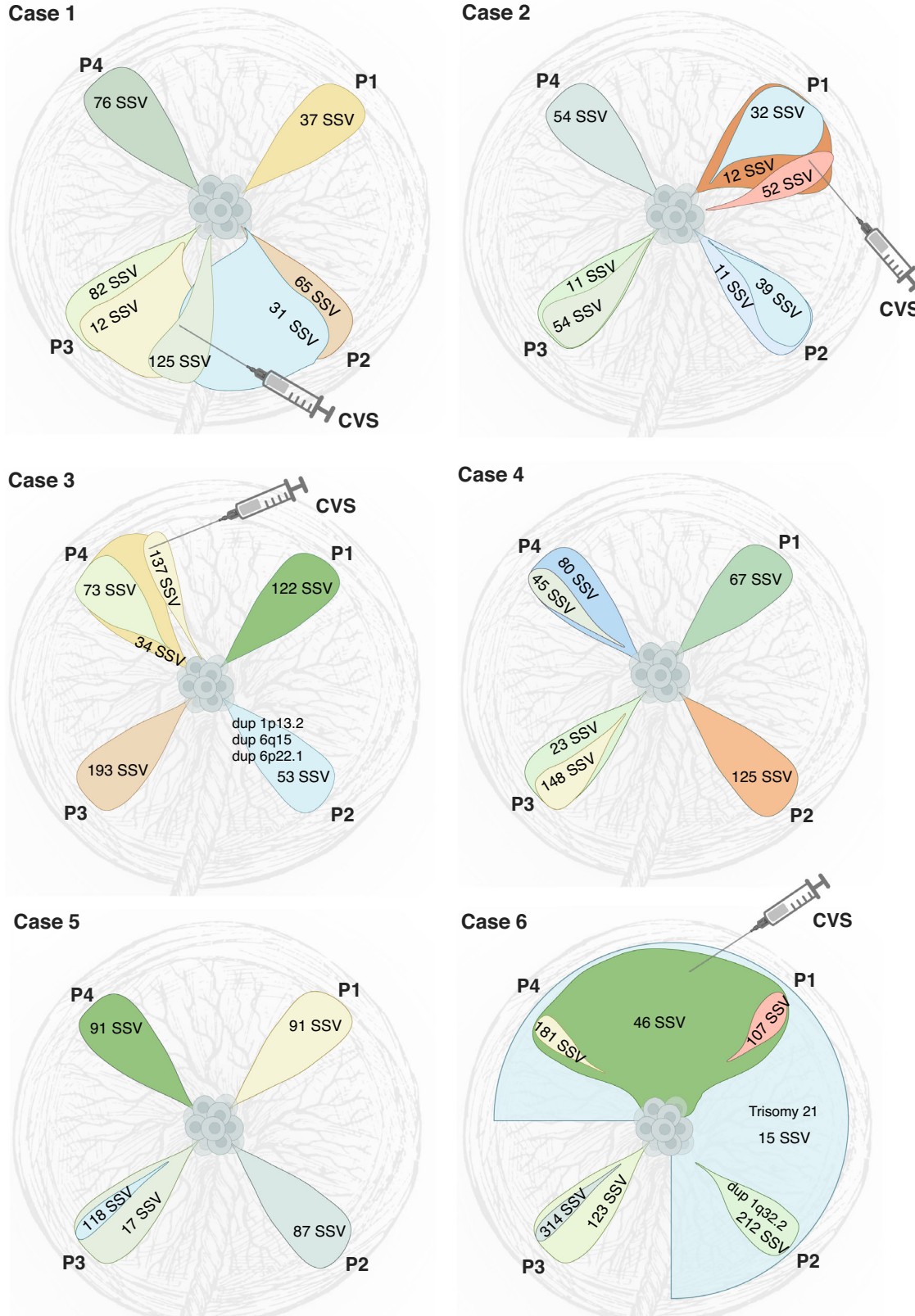

**Fig. 3 | Schematic representation of clonal evolution in placental biopsies for each case. Each drop-like insert illustrates the largest clonal expansion per biopsy, while subsequent clonal expansions are depicted relative to their size.** The number of postzygotic small sequence variants (PZVs) per clone, indicating the mutational burden, is annotated on each clone. Postzygotic copy number variants (CNVs) are included in their respective clones. The figure illustrates clonal overlaps between biopsies and offers predictions for CVS sampling locations, while highlighting the mutagenesis and clonal evolution occurring in the developing placenta. Created in BioRender. Larsen, M. (2025) https://BioRender.com/s5sgar7.

datasets, providing further independent support for the authenticity of the identified variants (Supplementary Fig. 6).

### Extensive sub-clonal branching is a common feature of placental development

A subset of placental biopsies contained two or more clones (Figs. 2 and 3). In the second trimester, developing placentas (cases 1–5), we observed two different clones in 9 out of 20 biopsies (45%), whereas in the term placenta (case 6), we observed three clones in three out of four biopsies and two clones in the fourth sample. We refer to the largest observed clonal expansion as the dominant clone, hypothesizing that smaller subclones evolved from the main clone that subsequently accumulated additional PZVs (Fig. 3). For cases 1–5, we can only speculate that one clone is dominant and has given rise to a subclone. However, without further data to confirm the origin of these clones, it remains plausible that they represent two distinct clones, each carrying a unique set of variants. For case 6, we can decipher the evolution based on the clonal expansion sizes per sample—the sum of the cell fractions of clones 6 and 7 is above 100% for P1 and P4 placental biopsies in this case (Fig. 2B). Thus, we are more certain that there is a dominant clone 6, and a subclone derived from it carrying PZVs from both 6 and 7 clusters. The presence of multiple PZV clusters demonstrates the coexistence of multiple clones. Although the results allow us to illustrate the dynamic clonal architecture and speculate on the development of the placenta, the data from this study do not allow us to definitively determine whether they are subclones arising from the dominant clone or distinct clones that had evolved in parallel and were coincidentally sampled together.

### Expansion of early clones across larger parts of the term placenta

While each placental biopsy contained at least one unique private clone, in the case of the term placenta (case 6), several biopsies additionally exhibited shared clones (Fig. 3). These shared clones demonstrated their expansion across larger parts of the placenta. In particular, biopsies from the term placenta (case 6) revealed that two clones were shared between multiple samples, showcasing the variability and complexity of clonal expansion across the placenta. One of the clones (clone 6) was observed in two out of four, and the other one (clone 7) was present in three out of four quadrant placental biopsies (Fig. 2B). Notably, both these clones were detected in CVS samples taken in the first trimester but were not identified in the AF analysis.

### Clonal compositions reveal the approximate CVS sampling site

In four of the six cases, we obtained CVS biopsies during the first trimester, which allowed us to compare the genetic heterogeneity of CVS and placental biopsies. In all cases, we identified postzygotic clones in the CVS sample. In cases 2, 3, and 6, two distinct clones were identified in each CVS sample, whereas in case 1, three unique cell clones were detected (Fig. 2B). In all cases, at least one of the clones from CVS was shared with a corresponding placental biopsy, suggesting a specific quadrant of the placenta from which the CVS sample most likely originated (Fig. 3 and Supplementary Fig. 1). Thus, the clonal composition of CVS samples revealed the approximate sampling sites within the placenta. This showcases the likely presence of additional clones in the placentas to those we could detect by four random quadrant placental biopsies. Given that our study only included one biopsy from each quadrant of the placenta per case, it is conceivable that the specific sites from which CVS were taken did not uniformly match our quadrant biopsy sites.

### Postzygotic copy number variants less common in the developing placenta

Postzygotic copy number variants (CNVs) were called in all placental biopsies. In two of the six cases, we detected postzygotic CNVs that

were not found in fetal or parental samples. In total, CNVs were detected in only 3 out of 35 clones identified (8.6%) (Fig. 3). In case 3, three CNVs were observed in one of the placental biopsies (P2)—a 187 kb duplication (1p13.2), a 173 kb duplication (6q15), and a 103 kb duplication (6p22.1)—that were not present in other placental biopsies or the fetus (Supplementary Fig. 2). All CNVs were detected at mosaic levels of 40–50%, most likely originating from clone 7 found in the P2 biopsy. In case 6, three out of four placental biopsies (P1, P2, and P4) showed aneuploidy of chromosome 21 at 50–60% mosaic level. Mosaic trisomy 21 was also detected in the CVS (at 50% mosaic level), but was not observed in the AF or the umbilical cord blood (Supplementary Fig. 2). The trisomy most likely originated from clone 7, which was exclusively found in the three placental biopsies and the CVS sample (Fig. 3). Furthermore, in one of these placental biopsies (P2), a 292 kb duplication (1q32.2, at 50% mosaic level) was detected, corresponding to the clone unique to that biopsy (clone 2) (Supplementary Fig. 2).

### Clonal expansions and mutational burden increase with gestational age

Our findings indicate that the size of clonal expansions in the placenta increases with the GA. For example, this is illustrated in Fig. 1 by the largest clonal expansions (51–61%) observed in the term placenta (case 6) and the smallest expansions (15–31%) in the placental biopsies taken after the termination of pregnancy in the early second trimester (case 1). We noticed great variability in clonal expansion size and mutational burden, which mainly seems to be related to the specific pregnancy and the individual sample. Mutational burden depicts the total number of PZVs detected in a clone, while a clonal expansion refers to the clonal cell fraction of the whole sample. Interestingly, in two out of four cases that underwent CVS procedure during pregnancy, the CVS samples exhibited larger clonal expansions than in matched placental biopsies (cases 2 and 3). In another one of those four cases (case 1), we observed a higher than the anticipated mutational burden in the CVS analysis (Fig. 2B). To minimize the effect of outliers or sample-to-sample variability, we assessed the mean largest clonal expansion and the mean mutational burden of four quadrant placental biopsies per case (Fig. 4A, B). This approach revealed a strong correlation between the sizes of clonal expansion and the GA ($R = 0.861$) (Fig. 4A). The same tendencies were observed for mutational burden, which demonstrated an increase with advancing pregnancy (Fig. 4B).

### Characteristics of fetal de novo germline variants

The number of fetal de novo germline SSVs found in the fetal tissue ranged from 52 to 72 SSV per case across the whole genome. Fetal de novo germline SSVs refer to variants present in the fetus but absent from parental samples. Specifically in the coding region, germline SSVs ranged from zero to two per case (Supplementary Table 1). For instance, out of the 54 germline SSVs across the genome in case 1, 58 in case 3, and 60 in case 4, only one SSV per case was detected in the coding region. Case 2 had a total of 72 germline SSVs with two in the coding region, and case 5 had no SSVs observed in the coding region. In case 6, de novo germline SSVs were excluded to facilitate PZV analysis due to the absence of a paternal sample for this case. Regarding the pathogenicity of the variants in the coding region, only one clinically relevant de novo germline SSV was identified in the investigated pregnancies. This heterozygous variant was an indel in the *NR2F2* gene identified in case 2 (Table 1) and has been reported to the patient as part of the clinical workflow[8]. Across all cases, de novo germline SSVs, including the clinically relevant indel in case 2, were consistently detected in each placental biopsy, as well as in the available CVS samples.

### No genetic differences observed between amniotic fluid and the fetus

For cases in which an AF sample was analyzed (cases 4–6), we did not observe any genetic differences between the fetus and the

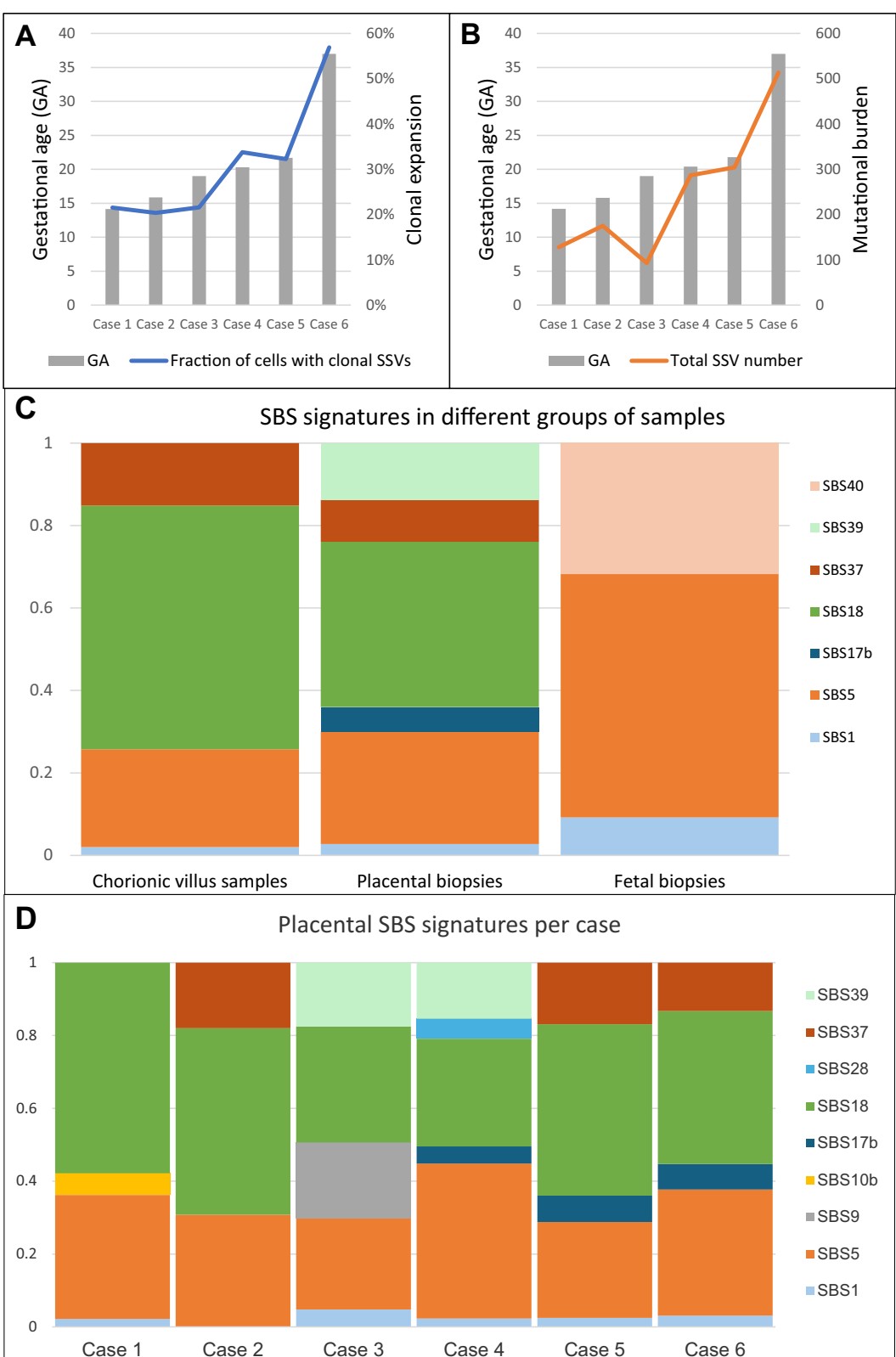

corresponding AF sample (Fig. 1). Interestingly, one minor private clone (clone 9) in case 4 was exclusively observed in the fetus and AF. It consisted of 22 PZVs and was observed in around 15% of the cells in both fetal and AF samples. This clone was not detected in any of the placental biopsies, hinting at the mosaicism confined to the fetus (Fig. 2B). It should be noted that in this study, the minor clonal

expansions that were observed in less than 10% of the cells per sample (i.e., low mosaic level), were excluded from our analysis. We set the lower detection limit for reliable detection of clonal events; however, clinically relevant low-frequency clones with <10% in the placenta may still exist at a higher level in specific fetal tissues in cases of true fetal mosaicism.

**Fig. 4 | Analysis of clonal expansions, mutational burden, and mutational signatures across gestational ages and sample types. A** The mean fraction of the largest clonal expansions observed across placental biopsies in each case, plotted by gestational age (GA). Larger clonal expansions were observed at later gestational ages, suggesting overall enlargement of clonal expansions over time. **B** The mean mutational burden across placental biopsies in each case, plotted by gestational age. Mutational burden is defined by the total number of postzygotic variants (PZVs) per sample. A positive correlation is observed between advancing gestational age and mutational burden, indicating continuous acquisition of new mutations as the placenta grows and develops. **C** Distribution of single-base

substitution (SBS) signatures across different sample types, including chorionic villus samples (CVS), postgestational placental biopsies and fetal tissue. CVS and placental biopsies include only PZVs, while fetal tissues (amniotic fluid or umbilical cord blood) include germline de novo variants. Notably, SBS18, linked to oxidative stress, was more pronounced in placental tissues, while germline-associated signatures, such as SBS1 and SBS5, dominated fetal samples. **D** Distribution of SBS mutational signatures across different cases. This analysis highlights the variability and similarities in mutational processes among the cases shaping placental genetic landscapes. Source data are provided as a Source data file.

## Mutational processes in early and late placenta

Mutational signatures of single-base substitutions (SBS) were developed during cancer studies; they represent mutational processes in different cancerous and noncancerous tissues[9]. We identified mutational signatures from the postzygotic substitution variants found in placental biopsies and CVS samples to explore the underlying mutational processes in the early and late placentas. Additionally, we characterized the mutational signatures of fetal de novo germline SSVs that occurred in gametes to explore potential differences in the mutational processes when compared to placental PZVs. The SBS signatures, SBS1 and SBS5, which are commonly found in noncancerous human tissues and reflect natural endogenous processes[10], were detected throughout all the samples: CVS, quadrant placental biopsies, and fetal samples (Fig. 4C). The fractions of SBS1 and SBS5 signatures are substantially higher among the de novo germline SSVs found in the fetal sample group (i.e., AF, Achilles tendon or umbilical cord), accounting for 9.2% and 59.02% of substitutions, respectively, compared to CVS or placental biopsies, where they appear at lower fractions (2.04% and 23.72% in CVS; 2.78% and 27.28% in placental biopsies, respectively). Since variants from SBS1 and SBS5 signatures are reported to be acquired with age[9], they are commonly observed in all healthy human tissues. SBS18 was discovered to be the most prominent signature across all included cases and demonstrated high prevalence in both CVS (59.1%) and placental biopsies (40.06%). SBS18 has previously been reported as a hallmark of the term placentas[7]. However, SBS18 was not identified in the de novo germline SSVs detected in the fetus. Additionally, the SBS37 signature, which has no reported etiology, was observed in both CVS and placental biopsies. In contrast, the SBS39 signature, also of unknown origin, was detected only in quadrant placental biopsies. Notably, neither SBS37 nor SBS39 signatures have previously been reported in placental tissues. Additionally, SBS40, another signature of unknown etiology[9], was exclusively observed in fetal samples and not present in any placental samples. When analyzing placental biopsies from each case separately, we observed additional case-specific SBS signatures of low frequency (Fig. 4D).

## Placental postzygotic variants in cell-free DNA from maternal plasma

As prenatal analysis based on cfDNA is advancing, we were inclined to examine whether PZVs from placental biopsies are detectable in cfDNA from maternal plasma. We had previously performed deep whole-exome sequencing of cfDNA extracted from maternal plasma samples[11] from the pregnancies involved in this study, enabling direct comparison of PZVs found in the exome region in placental samples with maternal plasma taken during pregnancies. Among the coding PZVs identified in placental biopsies, only a single variant from case 5 was detected in the maternal plasma sample (Supplementary Table 2). The other PZVs were not called in maternal plasma. Weak signals below the noise level were observed for a few of these placental PZVs. It is uncertain whether this represents true findings or sequencing artifacts, as these variants were only present in 1–2 reads. However, this data suggests that placental PZVs may not be present at readily

detectable levels in maternal plasma even when using advanced deep sequencing methods. While all coding fetal de novo germline SSVs were accurately detected in maternal plasma, as we reported previously[11].

## Discussion

The placenta is a complex organ that undergoes substantial genetic and epigenetic changes during development, and its heterogeneity can pose challenges for an accurate prenatal diagnosis[3,4,7]. Thus, the accuracy of prenatal diagnosis by CVS analysis is challenging because of the nature of placental development, as the placenta may exhibit genetic changes that are absent from the fetus and can lead to false results. Previous studies have predominantly examined large chromosomal aberrations found to be confined to the placenta. Recently, Coorens and colleagues reported extensive and ubiquitous mutagenesis in placental tissue from a cohort of term placentas analyzed by WGS[7]. In this study, we aimed to explore whether this phenomenon is already present at earlier stages of placental development, particularly at the time when placental biopsies are typically obtained for prenatal diagnostic purposes. We sought to delineate the dynamics of placental development from early pregnancy, offering additional insights needed to enhance the accuracy and reliability of genomic methods used in prenatal diagnostics. Our study advances this knowledge by employing deep genome sequencing to meticulously analyze early-stage placental development, an area that has not been comprehensively explored in previous research. By focusing on SSVs, including SNVs, indels variants, and CNVs, we provide unprecedented insights into the spatial and temporal heterogeneity of the placenta.

We demonstrate that extensive clonal expansions are already present during early placental development, with genetically distinct clones detected in every placental sample, including CVS specimens collected during ongoing pregnancies. Each clone was defined by a unique set of PZVs. Our results demonstrate that although clones comprising PZVs are ubiquitously present across all investigated placental samples, postzygotic CNV events are much rarer, identified in 8.6% of clones across all placental samples included in this study. By exploring temporal heterogeneity, we discovered that clones in the placenta develop and expand with GA, resulting in an increase in both the fraction of clonal cells and the mutational burden of the placental tissue. This variability underscores extensive clonal branching as a characteristic feature of placental development, suggesting an intricate collage of genetic variants across different placental regions.

Furthermore, our study confirmed extensive heterogeneity of the term placenta, with unique clones present across different quadrant placental biopsies per case, demonstrating clonal mosaicism at the sequence-variant level. Clonal expansions of cells with specific and unique sets of PZVs have been observed in the term placenta, demonstrating much more prevalent genetic heterogeneity compared to other organs and tissues[7]. If a gene crucial for placental function is hit by postzygotic mutations, these clonal expansions could affect the functioning of the placenta in both early and late pregnancy, potentially leading to adverse outcomes for the fetus, as observed with chromosomal-level mosaicism[4]. Thus, detailed exploration of

developing placental heterogeneity is crucial knowledge, which can aid in the development of prenatal screening and diagnostic methods, preventing misdiagnosis and improving clinical outcomes.

The integration of exome- and genome-based comprehensive sequencing into prenatal diagnostics has revealed that the major genetic cause of fetal structural anomalies is de novo variants[12,13]. From the results of this study, we conclude that de novo germline SSVs across the genome were detected with a 100% recall rate and high sensitivity in all analyzed placental samples, including early CVS samples. Out of 52–72 PZVs detected across the genome per case, only very few of them (0–2) were located in the protein-coding regions, and thus potentially of clinical relevance. These numbers align with what is already known from family-based genome studies[12].

Furthermore, we found that PZVs from placental clones were rarely detectable in cfDNA from maternal plasma by ultra-deep exome sequencing, whereas all analyzed germline SSVs were successfully identified (only variants in coding regions were analyzed in cfDNA) in our study. As each clone may comprise only a minor part of the total placenta, the amount of shed cfDNA from a single clone appears to be below the detection sensitivity level. Based on our results, we cannot definitively exclude the presence of PZVs in cfDNA from the maternal plasma, as they may be below the detection threshold of the test. These results provide additional insight for the developing field of comprehensive cfDNA-based analysis[14], suggesting a high detection accuracy of potentially clinically relevant fetal de novo SSVs while maintaining a low false positive rate.

Interestingly, we observed a higher level of clonal variability in direct CVS samples compared to placental quadrant biopsies of similar GA. We hypothesize that this may be attributed to differences in sampling technique and hence different cell type compositions, as well as natural development of the placenta. In two out of the four CVS samples, the CVS clone was found to be significantly more expanded than in the quadrant placental biopsies taken after pregnancy termination. This could indicate that these CVS samples may contain a higher proportion of trophoblastic cells, as it has previously been described that this cell type contains higher levels of clonal expansion than mesenchymal cells[7]. Clonal expansions identified in CVS might not expand proportionately as the placenta continues to develop, and thus may be undetectable in later-stage biopsies. While it was not a uniform finding, in some cases, however, there was a clonal overlap observed between CVS and later-stage placental biopsies, indicating a likely CVS sampling location. This suggests that CVS sampling may not consistently align with late placental biopsies, highlighting the potential challenges in full placental genetic heterogeneity representation based on a single CVS, as it was previously reported in terms of chromosomal aberrations[15]. As some of the early events captured by CVS may be pushed out and overlooked in subsequent placental development, it also provides insight into how well an early placental biopsy (CVS) represents the final placental genetic architecture. While most placental PZVs may not be clinically relevant as they are restricted to noncoding regions, it is important to differentiate between fetal variants and confined placental variants when analyzing CVS sequencing data in the clinic, particularly taking caution when no corresponding fetal phenotype is observed, thus no a priori risk for a genetic condition, as the majority of monogenic diseases do not show phenotype prenatally[16]. In our study, all fetal de novo germline SSVs were fully present in the placenta, implying that they would not be missed by the CVS analysis. In contrast to placental biopsies, AF samples showed much less heterogeneity and fully represented the fetal genetic variation landscape. Thus, in cases of a clinically significant variant, especially presenting at a higher VAF, confirmatory test using AF would be deemed essential. Overall, our study supports the need to be aware of the risks of CPM and to follow up with amniocentesis whenever relevant. Our study supports amniocentesis to be the gold standard when CPM is suspected[4].

When analyzing the SBS mutational signatures, we identified several prominent signatures across different sample types. As previously reported to be associated with age-related endogenous processes[10], SBS1 and SBS5 were the most common among all samples, although they were more prevalent in fetal than in placental samples. Additionally, SBS18, which is linked to the cell damage caused by reactive oxygen species and had been previously observed in term placentas[7,10], was also present in all placental samples (including CVS) but absent in fetal samples, hinting towards the mechanism indicating the postzygotic origin of the associated variants. The presence of the SBS18 signature in the placenta suggests that oxidative stress plays a significant role in placental biology and is most likely attributed to its high metabolic activity. Finally, we extracted some mutational signatures of unknown etiology[9], which, to our knowledge, have not been previously reported in placental or fetal tissue analysis. The exclusive detection of SBS37 and SBS39 in placental samples and SBS40 in fetal samples underscores the distinct mutational architecture of placental and fetal tissues.

In conclusion, this study increases our knowledge of the genetic architecture of the developing placenta. We demonstrated that postzygotic de novo SSVs are present in both non-coding and coding regions of the placenta and exhibit extensive heterogeneity across different locations and time points during pregnancy. Our findings suggest that certain clones in the placenta may be more widespread than others because of early mutations or advantageous growth. We additionally confirmed that due to clonal similarity to fetal tissue, AF is the optimal sample when a clinically relevant mosaic variant is suspected. While the number of PZVs in the placenta is large, only a few of these variants are in the coding region and could potentially be relevant in clinical diagnostics. When using comprehensive sequencing technologies, it is important to be aware of placental heterogeneity in the prenatal diagnostics setting. Despite extensive clonality, our analysis indicates that the majority of placental PZVs are not clinically relevant. Additionally, placental PZVs are often easily discernible, as they have lower VAFs observed in the sequencing data analysis. Importantly, no false negatives for fetal de novo germline SSVs were observed in our study, suggesting high reliability in detecting true fetal variants from placental samples and maternal plasma cfDNA. There are some limitations of this study due to a small sample size, and the samples all being from pregnancies with ultrasound abnormalities. Future studies of a larger number of cases, a larger number of samples from various GAs, and also including normal pregnancies, are needed to gain further insight into the evolution and branching of clonal expansions during placental development. Despite these limitations, our study contributes important insights to the field. It provides a more comprehensive understanding of the spatial and temporal genetic heterogeneity of the developing placenta, emphasizing the need for careful consideration of sample type and timing in prenatal diagnostics.

## Methods

### Subject inclusion and sample collection

This study was approved by the Research Ethics Committee of the Region of Southern Denmark (Project-ID: S-20190027). Six pregnant individuals (age range: 25–33 years) were enrolled at the Department of Clinical Genetics and the Department of Obstetrics and Gynecology, Odense University Hospital, Denmark, all of whom had indications for prenatal genetic testing, and causative chromosomal or sequence variants were previously identified for most of them and reported in a clinical setting (Table 1). The inclusion of pregnancies with detected abnormalities enabled the collection and comprehensive analysis of placental and fetal samples at various gestational stages. This study complies with ethical principles outlined in the Declaration of Helsinki, which stands against performing unnecessary invasive and risk-inducing procedures solely for research in otherwise healthy

pregnancies. Informed consent was signed and obtained from all participants prior to inclusion. No participant compensation was provided, in accordance with institutional and ethical guidelines.

Biological sex of the fetal and parental samples was determined from genetic data (Supplementary Table 1). The sex of pregnant individuals was assigned based on medical records and genetic data as female; no additional gender identity data were collected. The study did not involve sex- or gender-specific research questions or stratified analyses, as the primary aim was to investigate placental mosaicism and postzygotic variant distribution in a limited sample cohort.

For each case, four random postgestational placental biopsies, that is, after termination of pregnancy or after birth, were collected from different quadrants of the placenta to investigate the genetic heterogeneity of the placenta. Additionally, fetus-representing tissue, that is, Achilles tendon or umbilical cord blood, as well as parental peripheral blood samples, were obtained to aid in variant calling and comparison. Prenatal samples, i.e., AF of 15–20 mL or CVSs of 30–50 mg, were obtained for each case during pregnancy to analyze any differences between the prenatal samples and the fetus. A graphical illustration of the samples obtained per case with the corresponding GA is depicted in Fig. 1.

### DNA extraction
Genomic DNA was extracted from placental and Achilles tendon biopsies using the QIAamp DNA Mini Kit (Qiagen; cat. #51304) following the manufacturer's protocol. DNA from AF and peripheral blood leukocytes was extracted using a Maxwell RSC Cell DNA Purification Kit (Promega; cat. #AS1370). Qubit dsDNA BR Assay Kit (Qubit; cat. #Q32850) was used to assess DNA concentration and purity.

### Genome sequencing
Genome sequencing was performed using an Illumina DNA PCR-Free Prep Kit (Illumina, cat. # 20041794). Library preparation was performed according to the manufacturer's instructions using IDT for Illumina DNA/RNA unique dual indexes (Integrated DNA Technologies; cat. #20027213) with 1 μg of genomic DNA as an input. Libraries were sequenced using the NovaSeq 6000 platform (Illumina, Cat.#2001285). All fetal and prenatal samples, including placenta biopsies, AF, umbilical cord, and Achilles tendon, were sequenced with a mean depth of 146× (range: 100×–214×) and parental samples were sequenced with a mean depth of 46× (range: 36×–56×).

### Alignment and variant calling pipeline
Sequencing data were processed using the DRAGEN Bio-IT Platform v4.03 (Illumina) using the DNA pipeline in somatic single-sample mode. Raw reads were aligned to the Illumina DRAGEN Graph Reference Genome GRCh38. For SNV/INDEL variant analysis, systematic noise filtering was applied using a custom Panel of Normals (PoN) constructed from 46 unrelated whole-genome sequencing (WGS) reference samples processed through the same pipeline. This PoN was used to empirically identify site-specific noise and recurrent artefacts, enabling the construction of a noise model that was subsequently applied to suppress false positive variant calls. Variant calls from each sample from the same pregnancy were merged into one vcf-file using BEDtools 2.31.1. The merged vcf-files were thereafter used for forced variant calling using the DRAGEN DNA somatic pipeline of all related fetal and parental samples. Variant annotation, filtering and visualization were carried out using VarSeq v2.3.0 (Golden Helix). A variant was considered postzygotic (somatic) in placenta tissue, when meeting the following filtering criteria in one of the placenta samples: Variant allele frequency (VAF) > 5%, and SQ (somatic quality) >10 in placenta sample and VAF < 3% or >20× coverage in matched fetal samples and maternal and paternal blood samples. A variant was considered fetal de novo germline when meeting the following filtering criteria: VAF > 5%, and

SQ (somatic quality) >10 in fetal samples, and AF < 3% or >20× coverage in matched maternal and paternal blood samples. Only variants in regions with sequencing coverage >50× in all related placenta and fetal samples were considered. Variant calls were restricted to high-confidence regions of the genome. Specifically, low-complexity regions and areas of low mappability were excluded using stratification BED files provided by the Genome in a Bottle (GIAB) consortium (see "Data availability"). To further minimize recurrent technical artefacts, variants observed in >10 individuals in the gnomAD database or in more than 10% of a local cohort ($n = 1300$ whole-genome samples) were removed to avoid recurrent artefacts.

Filtering criteria were defined based on DRAGEN somatic pipeline evaluation study by Scheffler et al.[17]. Scheffler and colleagues demonstrated a high detection rate and precision for both SNVs and indels, with a very low false positive rate for somatic variants with VAF > 5% in deep sequenced genome samples (>80× mean coverage). Based on these findings, a conservative cutoff of 5% VAF was selected, even though the study samples were sequenced at a higher depth. This threshold was chosen to ensure high-confidence variant calls, minimize false positives, clearly distinguishing true somatic variants from sequencing artifacts (Supplementary Fig. 3). To further validate the filtering approach, random variants were selected and manually curated through visual inspection and evaluation of the aligned reads. This manual curation process helped assess the reliability of variant calls and confirm the effectiveness of the filtering criteria in distinguishing true variants from sequencing artifacts (Supplementary Fig. 4).

Postzygotic copy number analysis was called using the DRAGEN DNA pipeline's sub-workflow for Somatic WGS CNV calling. Only CNV calls spanning regions greater than 50 kb and copy number <1.9 or >2.1 in at last one placenta sample were considered with no call in matched fetal samples and maternal and paternal blood samples. For Case 6, paternal blood samples were not available, here only fetal sample were used in the comparison.

### Inferring and mapping the clonal populations in the placenta
PyClone v0.13.1 software (https://github.com/Roth-Lab/pyclone) was used to identify and quantify clonal populations in the placental samples. Lists of postzygotic substitution variants for each placenta sample, were used as input, to recognize the cellular clonal cell populations. PyClone outputs clusters of variants based on calculated cellular frequencies. Variant clusters are subsequently used to characterize the clonal heterogeneity of the placenta and identify unique and shared cell clones present in placental biopsies from different quadrants and prenatal samples. For visualizing clusters of postzygotic variants and their intersample and interclonal relationships, an R-based ShinyApp software was created (https://mjlarsen.shinyapps.io/VafVaf-v0-4).

### Analyses of postzygotic placenta mutations in cell-free DNA from maternal plasma
To investigate the detection of postzygotic placental mutations in maternal plasma cfDNA, we analyzed whether PZVs identified in placental biopsies are present in maternal plasma. We had previously performed deep WES (mean, ~4000×) on cfDNA extracted from maternal plasma samples from the same pregnancies[11]. Maternal peripheral blood (8 mL) was collected in Streck Cell-Free DNABCT tubes (Streck; cat. #218996). Plasma was isolated and cfDNA extracted using the Cell3 Xtract kit (Nonacus Ltd; cat. #PRE_EXT_C3X_16). Sequencing libraries were prepared with the Cell3 Target: Exome CG kit (Nonacus Ltd; cat. #C3321EX) and multiplexed (8–12 samples per run) for sequencing on an Illumina NovaSeq 6000 (S4 flow cell; cat. #20028312). Fetal fractions in the analyzed plasma samples ranged from 3.72 to 19%. PZVs identified in the exome regions of placental samples were compared against deep exome sequencing data of matched maternal plasma samples collected during pregnancy. The

presence of these variants in maternal plasma was assessed by identifying supporting reads and VAFs indicative of postzygotic placental variants.

## Mutational signature analysis

SBS signature analysis of postzygotic placenta variants and fetal de novo germline variants was conducted using COSMIC mutational signature profiler SigProfiler (https://cancer.sanger.ac.uk/signatures). Only substitution variants were included in the SBS signature analysis.

## Verification of postzygotic placenta mutations by deep targeted sequencing

Deep targeted sequencing was performed on the P3 sample from Case 2 to verify the presence of candidate postzygotic mutations in the placenta. Genomic DNA libraries were prepared using the Ligation Sequencing kit V14 (Oxford Nanopore Technologies; cat. # SQK-LSK114) and sequenced on R10.4.1 flow cells using the PromethION P24 platform (Oxford Nanopore Technologies; cat. #FLO-PRO114M) with adaptive sampling to enrich for target regions. The regions of interest (ROIs) comprised 254 loci corresponding to mutation sites previously identified in Case 2 samples. Basecalling was performed using the SUP model in Dorado v0.9.6, and reads were aligned to the reference genome using minimap2. Read counts at all 254 variant positions were extracted using BCFtools v1.22 with quality filters set at -q 20 and -Q20.

## Reporting summary

Further information on research design is available in the Nature Portfolio Reporting Summary linked to this article.

## Data availability

The processed small somatic variant (SSVs) calls generated from genome sequencing of the study samples have been deposited and are publicly available in the European Variation Archive (EVA) at EMBL-EBI under accession number PRJEB95774. The raw sequencing data from deep WGS analysis cannot be shared publicly or under confidentiality agreements due to legal and ethical restrictions governed by the General Data Protection Regulation and Danish national legislation, including the Danish Act on Processing of Personal Data. Specifically, these datasets contain identifiable genomic germline information from fetal and parental samples, as well as the somatic placental variants described in the manuscript, and the ethical approvals obtained from the Danish National Committee on Health Research Ethics and local Institutional Review Boards did not include consent for germline data sharing beyond the original research team. As such, this restriction prohibits any third-party sharing, including under data-use or confidentiality agreements. Areas of low mappability were excluded using the stratification BED files provided by the GIAB consortium: https://ftp-trace.ncbi.nlm.nih.gov/ReferenceSamples/giab/release/genome-stratifications/v3.1/GRCh38/mappability/GRCh38_lowmappabilityall.bed.gz. The raw sequencing data from deep WES analysis of cfDNA from our previous study (ref. 11), used in the current study for the analysis of postzygotic placental variants in maternal plasma, cannot be shared publicly or under confidentiality agreements for the same reasons described above. These data contain identifiable genetic information, and the ethical approvals obtained for this study did not include consent for sharing the data beyond the original research team. Source data for figures and tables are provided with this paper.

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

## Acknowledgements

We are deeply grateful to the study participants for their invaluable contributions. We thank the laboratory technicians at the Department of Clinical Genetics, Odense University Hospital, for their expert technical assistance, and the staff at the Department of Obstetrics and Gynecology, Odense University Hospital, for their support with sample collection. This study was supported by grants from the Region of Southern Denmark (18/17848, I.M., 20/14085, C.F. and 21/17619, M.J.L.) and the A.P. Møller Foundation for the Advancement of Medical Science (19-L-0370, I.M.).

## Author contributions

I.M. co-designed the study, performed the analyses, interpreted the results, and wrote the manuscript. M.J.L. co-designed the study, supervised the analysis and interpretation, and contributed to writing. C.F., C.B.-A., and P.M.T. co-designed the study and contributed to result interpretation and discussion. C.F., L.S., B.S.K., and P.M.T. coordinated sample acquisition and provided clinical insights. Q.H. co-designed and performed bioinformatic processing, variant filtering and interpretation. All authors discussed the results, contributed to manuscript revisions and approved the final version.

## Competing interests

The authors declare no competing interests.

## Additional information

Ieva Miceikaite or Martin J. Larsen.

