## [Transparent Peer Review file · Nature Communications]

Deep Genome Sequencing Reveals Extensive Genetic Heterogeneity in Early Human Placentas

Corresponding Author: Dr Ieva Miceikaite

Version 0:

Reviewer comments:

Reviewer #1

(Remarks to the Author)

This well written manuscript provides information regarding placental mosaicism for small nucleotide variant and small indel variants at different gestational stages. Although there is a wealth of data on confirmed placental mosaicism for chromosomal aberrations there is very little on small sequence variants. This study is therefore of great interest albeit with a small sample number. There are some areas that need addressing:

Lines 395-400 Setting for thresholds for calling mosaicism: Please provide clarity on how these thresholds were determined. It is a challenge to differentiate real mosaic variants against the background of sequencing artefacts. Were there audits performed with variants that were known to be real confirmed by another validated method such as digital PCR? Further data supporting these thresholds in the supplementary data would be useful.

Please add in VAF for sample type where variant was detected in supplementary table 2. If lower than 50% (as in fetal de novo) could the lack of detection in cell free DNA be due to the lower level VAF and the reason why the variants are not being seen / in too few reads to be called? Please also include the fetal fraction as this will also impact whether these variants are observed in cfDNA.

Following this, the statement on line 232: "However, it demonstrated that placental PZVs are not present at readily detectable levels in maternal plasma even using advanced deep sequencing methods." This is strongly worded considering there is only data from 5 samples and a small number of coding variants and will be dependent on fetal fraction. "The data suggests" may be better wording along with suggestion of further studies. This applied to lines 292-295 in the discussion as well.

The authors have included discussion about testing amniotic fluid for confirmation where a clinically mosaic variant is suspected. However, please add discussion on the implications on identifying "de novo" variants in CVS samples in fetuses with no matching phenotype. There is a growing amount of recent literature suggesting diagnostic utility of exome sequencing in structurally normal fetuses using invasive samples some CVS samples. Also, there are the well-publicised issues of incidental findings. Does your data support that all variants that are de novo in CVS samples with no detectable phenotype warrant confirmation in AF samples or does it depend on the VAF?

Although briefly mentioned in the conclusions the limitations of the small sample number particularly considering the range of gestations needs to be highlighted more in the discussion.

Minor comments:

Line 99 – "we observed tree" should read "we observed three"

Line 419 – incomplete sentence "In some cases"

Reviewer #2

(Remarks to the Author)

I have reviewed the manuscript entitled 'Deep Genome Sequencing Uncovers Extensive Genetic Heterogeneity in Early Human Placentas'. This manuscript is novel as the need for awareness and caution when using placental biopsies as fetal proxy for diagnostics and emphasize the importance of confirmatory testing using amniotic fluid when placenta mosaicism is suspected. However, major changes should be done in order to make the manuscript understood. Some suggestions are as follows:

Major comments

It is difficult to generalize this study since there were no cases without fetal abnormalities among the cases reviewed. It is also difficult to know whether conclusions can really be drawn from only six cases.

Minor comments

Introduction

Line 46: The authors should mention in the introduction that human placental time series studies are difficult to obtain specimens

Results

Line 77 : The authors should give a more detailed explanation of Figure 1 and its interpretation.

ONLINE METHODS

Line 367: The authors should clarify the amount of amniotic fluid and chorionic villi collected

Line 424 : The authors should not only raise references, but also describe the methodology of analyses of postzygotic placenta in cell-free DNA from maternal plasma

Figures and Tables

Line 464: The authors should give a more detailed explanation of Figure 4 and its interpretation.

Version 1:

Reviewer comments:

Reviewer #1

(Remarks to the Author)

Thank you for the amendments to the manuscript, it is much improved. However, there are still issues to be addressed with how these variants are determined to be real and not false positive sequencing artefacts.

The article that you have referenced in response to comment 1 (doi: <https://doi.org/10.1101/2023.03.23.534011>) has not been peer-reviewed and is from a company that sells software that calls mosaic variants. It is therefore highly likely that this article is biased. I appreciate that pile up/alignment images have been provided however some of these do show messy sequence (clone 5.1 & 5.2). In this reviewer's experience you can get false positive variants with VAFs over 5% and therefore more evidence needs to be provided to confirm that these mosaic variants are real and are not sequencing artefacts - ideally using another technique such as ddPCR or Sanger sequencing (if the VAF is over 10%).

Reviewer #2

(Remarks to the Author)

I have reviewed the revised manuscript entitled 'Deep Genome Sequencing Reveals Extensive Genetic Heterogeneity in Early Human Placentas' This manuscript is novel as the need for awareness and caution when using placental biopsies as fetal proxy for diagnostics and emphasize the importance of confirmatory testing using amniotic fluid when placenta mosaicism is suspected. I confirmed that the manuscript has been changed according to the reviewer's comments.

Version 2:

Reviewer comments:

Reviewer #1

(Remarks to the Author)

Many thanks for the additional information that provides verification that the pipeline is calling variants that are truly mosaic rather than artefacts. It is appreciated that the authors are unable to confirm all variants and therefore the manuscript is now suitable for publication.

Authors' Response Letter

Dear Editors and Reviewers,

We sincerely thank the reviewers for their thoughtful and constructive feedback, which have enhanced the quality of our manuscript.

Below, we provide a point-by-point response to the reviewers' comments, outlining the changes made in the revised manuscript. Responses are provided below in *blue cursive*.

Reviewer #1 (Remarks to the Author):

This well written manuscript provides information regarding placental mosaicism for small nucleotide variant and small indel variants at different gestational stages. Although there is a wealth of data on confirmed placental mosaicism for chromosomal aberrations there is very little on small sequence variants. This study is therefore of great interest albeit with a small sample number. There are some areas that need addressing:

Lines 395-400 Setting for thresholds for calling mosaicism: Please provide clarity on how these thresholds were determined. It is a challenge to differentiate real mosaic variants against the background of sequencing artefacts. Were there audits performed with variants that were known to be real confirmed by another validated method such as digital PCR? Further data supporting these thresholds in the supplementary data would be useful.

A: Thank you for your thoughtful comments and for recognizing the significance of our study. We appreciate the opportunity to clarify and expand on the points you raised.

Regarding the thresholds for calling somatic variants, filtering criteria were defined based on the DRAGEN somatic pipeline evaluation study by Scheffler et al., 2023 (ref, <https://doi.org/10.1101/2023.03.23.534011>), along with extensive experience in somatic variant detection from multiple cancer studies investigating clonality. Scheffler and colleagues demonstrated an ultra-high detection rate and precision, with a near 100% detection rate and a very low false positive rate for somatic variants with a variant allele fraction (VAF) >5% in deep-sequenced genome samples (>80× mean coverage). Given these findings, we selected a conservative cutoff of 5% VAF, even though our study samples were sequenced at a higher depth. This threshold was chosen to ensure high-confidence variant calls, minimize false positives, and avoid the need for additional confirmatory testing while clearly distinguishing true somatic variants from sequencing artifacts. To further validate the filtering approach, random variants were selected and manually curated through visual inspection and evaluation of the aligned reads. This manual curation process helped assess the reliability of variant calls and confirm the effectiveness of the filtering criteria in distinguishing true variants from sequencing artifacts. Supporting data for this decision is now provided in Supplementary Figures 3 and 4, and we have expanded the Methods section on Alignment and variant calling pipeline, accordingly to clarify this approach.

Please add in VAF for sample type where variant was detected in supplementary table 2. If lower than 50% (as in fetal de novo) could the lack of detection in cell free DNA be due to the lower level VAF and the reason why the variants are not being seen / in too few reads to be called? Please also include the fetal fraction as this will also impact whether these variants are observed in cfDNA.

A: We have updated Supplementary Table 2 to include VAF values in samples that variant was detected, and fetal fractions for each cfDNA sample. We have acknowledged the point that lower VAFs (as usually seen in postzygotic variants) may influence the detectability of variants in cfDNA as biologically lower amount of cfDNA fragments harboring those variants would end up in maternal

blood stream, requiring very high read depth to detect. The same would be the case for clonal variants only present in one part of the placenta. In our study we used ultra deep exome sequencing (read depth range: 2710x to 8075x, mean: 4548x) of cfDNA from plasma samples to enhance the ability to detect even ultra rare variants in cfDNA. Another limitation to consider is that the absolute count of circulating DNA fragments originating from the placenta is inherently dependent on the total amount of cfDNA present in the plasma sample. This dependency limits the overall complexity and may further impact the detection sensitivity for low-abundance placental variants.

Following this, the statement on line 232: "However, it demonstrated that placental PZVs are not present at readily detectable levels in maternal plasma even using advanced deep sequencing methods." This is strongly worded considering there is only data from 5 samples and a small number of coding variants and will be dependent on fetal fraction. "The data suggests" may be better wording along with suggestion of further studies. This applied to lines 292-295 in the discussion as well.

A: The noted changes have been applied to reflect this limitation, by lightening the statements in Results and Discussion sections.

The authors have included discussion about testing amniotic fluid for confirmation where a clinically mosaic variant is suspected. However, please add discussion on the implications on identifying "de novo" variants in CVS samples in fetuses with no matching phenotype. There is a growing amount of recent literature suggesting diagnostic utility of exome sequencing in structurally normal fetuses using invasive samples some CVS samples. Also, there are the well-publicised issues of incidental findings. Does your data support that all variants that are de novo in CVS samples with no detectable phenotype warrant confirmation in AF samples or does it depend on the VAF? Although briefly mentioned in the conclusions the limitations of the small sample number particularly considering the range of gestations needs to be highlighted more in the discussion.

A: We thank the reviewer for this valuable suggestion. We have expanded on variants in CVS samples with no corresponding fetal phenotype detected, and emphasize that our findings suggest a cautious approach to interpreting such variants, especially those with low VAFs, as they may represent confined placental mosaicism rather than fetal mutations. We highlight that in case of higher VAF and no phenotype, confirmation via amniotic fluid testing is critical as many monogenic disorders do not show phenotype prenatally.

We have expanded our limitation and future perspectives section. We now have emphasized that while our findings provide important insights, studies with a broader range of cases and higher number of samples are needed to validate and generalize these results.

Minor comments:

Line 99 – "we observed tree" should read "we observed three"

A: We have fixed the noted spelling error.

Line 419 – incomplete sentence "In some cases"

A: The incomplete sentence has been removed.

Reviewer #2 (Remarks to the Author):

I have reviewed the manuscript entitled 'Deep Genome Sequencing Uncovers Extensive Genetic Heterogeneity in Early Human Placentas'. This manuscript is novel as the need for awareness and caution when using placental biopsies as fetal proxy for diagnostics and emphasize the importance of confirmatory testing using amniotic fluid when placenta mosaicism is suspected. However, major

changes should be done in order to make the manuscript understood. Some suggestions are as follows:

Major comments

It is difficult to generalize this study since there were no cases without fetal abnormalities among the cases reviewed. It is also difficult to know whether conclusions can really be drawn from only six cases.

A: We acknowledge that the inclusion of only pregnancies with fetal anomalies limits generalizability. The purpose of our study was to add basic knowledge of the landscape of sequence variants in placental tissue. While our sample size is small, the inclusion of multiple biopsies and fetal samples per case provided robust insights into placental heterogeneity. We have included the reasons for the inclusion of pregnancies with fetal anomalies in the Online Methods section, and further emphasized these limitations and the need for larger, broader studies in the Discussion section.

Minor comments

Introduction

Line 46: The authors should mention in the introduction that human placental time series studies are difficult to obtain specimens

A: Thank you for highlighting this important point. We have updated the Introduction to acknowledge the challenges of obtaining sequential human placental samples, reflecting the lack of studies in this area.

Results

Line 77 : The authors should give a more detailed explanation of Figure 1 and its interpretation.

A: We have revised the Results section to provide a more detailed explanation of Figure 1 and its purpose.

ONLINE METHODS

Line 367: The authors should clarify the amount of amniotic fluid and chorionic villi collected

A: We have included the volume of amniotic fluid and the weight of chorionic villi samples collected in the revised Online Methods section.

Line 424: The authors should not only raise references, but also describe the methodology of analyses of postzygotic placenta in cell-free DNA from maternal plasma

A: We have expanded on this in the Online Methods section to describe the analytical pipeline for identifying postzygotic placenta variants in cfDNA, including quality control and variant calling criteria.

Figures and Tables

Line 464: The authors should give a more detailed explanation of Figure 4 and its interpretation.

A: We have included a more elaborate explanation of Figure 4 in the legend.

Thank you for the opportunity to revise and improve our manuscript. We look forward to your feedback on the revised version.

*Sincerely,
On behalf of all authors,
Ieva Miceikaitė, PhD
Postdoctoral Research Fellow, Center for Genomic Medicine
Massachusetts General Hospital*

Authors' Response Letter

Manuscript number: NCOMMS-24-67043A

We sincerely thank the reviewers and editors for their time and thoughtful feedback. We have revised the manuscript accordingly and believe the new version addresses all remaining concerns. Below, we provide a detailed point-by-point response to each comment, outlining the changes made and the additional analyses conducted. All modifications are reflected in the revised manuscript and/or supplementary materials.

Reviewer #1 (Remarks to the Author):

Thank you for the amendments to the manuscript, it is much improved. However, there are still issues to be addressed with how these variants are determined to be real and not false positive sequencing artefacts.

The article that you have referenced in response to comment 1 (doi: <https://doi.org/10.1101/2023.03.23.534011>) has not been peer-reviewed and is from a company that sells software that calls mosaic variants. It is therefore highly likely that this article is biased. I appreciate that pile up/alignment images have been provided however some of these do show messy sequence (clone 5.1 & 5.2). In this reviewer's experience you can get false positive variants with VAFs over 5% and therefore more evidence needs to be provided to confirm that these mosaic variants are real and are not sequencing artefacts - ideally using another technique such as ddPCR or Sanger sequencing (if the VAF is over 10%).

Response:

We appreciate the reviewer's emphasis on the importance of robust validation. To address this concern, we have taken the following steps:

- **Independent Data Confirmation:** We analyzed both clinical and research whole-genome sequencing datasets for CVS samples from Cases 1–3. Despite being prepared and sequenced independently (clinical standard WGS vs. research deep WGS), we consistently observed the same somatic variant clusters across both datasets (see Supplementary Figure 6). This cross-validation across separate workflows strengthens confidence in the authenticity of the identified variants. We have added Supplementary Figure 6 comparing variant detection between the two independent sequencing datasets (standard WGS vs. deep WGS), highlighting consistent detection of the same variants across sample prep and sequencing conditions.
- **Population-Level Filtering:** We screened a large cohort of ~1,300 unrelated genomes processed using the same DRAGEN pipeline. None of the somatic variants reported in our study were recurrent, supporting the conclusion that none of the variants are systemic artefacts, thus reaffirming their validity.
- **Nanopore Targeted Validation:** To provide orthogonal support, we performed deep targeted sequencing using Oxford Nanopore Technology and adaptive sampling for a selected example case (Case 2, placental sample P3). This allowed us to re-sequence

representative variants at very high depth using a different sequencing platform and chemistry. The variants were confirmed, supporting the reliability of our calls (see Supplementary Figure 5).

- **Clarification in Text:** We revised the manuscript to provide clearer descriptions of these validation steps in both the Methods and Results sections. We also refined the variant filtering language to highlight the rigorous measures taken to suppress noise and prevent artefactual calls.

While we acknowledge the value of orthogonal validation, the large number of variants and their often-low VAFs present practical challenges for techniques like Sanger sequencing or ddPCR. Our combined use of deep sequencing, independent datasets, and cross-platform validation offers a robust alternative, consistent with the approach used by Coorens et al. (Nature, 2021), whose findings were accepted without orthogonal validation.

Reviewer #2 (Remarks to the Author):

I have reviewed the revised manuscript entitled 'Deep Genome Sequencing Reveals Extensive Genetic Heterogeneity in Early Human Placentas' This manuscript is novel as the need for awareness and caution when using placental biopsies as fetal proxy for diagnostics and emphasize the importance of confirmatory testing using amniotic fluid when placenta mosaicism is suspected. I confirmed that the manuscript has been changed according to the reviewer's comments.

Response: We thank the reviewer for their careful review and kind words. We are pleased that the revised manuscript meets expectations, and we appreciate your support.

We hope that these comprehensive revisions and additional validation efforts address the remaining concerns and that the manuscript is now suitable for publication in Nature Communications.

Sincerely,

Ieva Miceikaitė, PhD

Postdoctoral Research Fellow, Center for Genomic Medicine, Massachusetts General Hospital
